Deep learning-based fine-grained assessment of aneurysm wall characteristics using 4D-CT angiography

Kumrai Teerawat teerawat.kumrai@ist.osaka-u.ac.jp 1
Maekawa Takuya maekawa@ist.osaka-u.ac.jp 1
Chen Yixuan 1
Sugiyama Yoshie 1
Takagaki Masatoshi 2
Yamashiro Shigeo 3
Takizawa Katsumi 4
Ichinose Tsutomu 5
Ishida Fujimaro 6
Kishima Haruhiko 2
1 Graduate School of Information Science and Technology, Osaka University , Suita , Osaka , Japan
2 Department of Neurosurgery, Osaka University Graduate School of Medicine , Suita , Osaka , Japan
3 Division of Neurosurgery, Department of Cerebrovascular Medicine and Surgery, Saiseikai Kumamoto Hospital , Kumamoto , Japan
4 Department of Neurosurgery, Japanese Red Cross Asahikawa Hospital , Asahikawa , Hokkaido , Japan
5 Department of Neurosurgery, Osaka Metropolitan University , Abeno , Osaka , Japan
6 Department of Neurosurgery, Mie Chuo Medical Center, National Hospital Organization , Tsu , Mie , Japan
Parker Matthew
Electronic publication date: 2025 May 9
Publication date: 2025
Volume: 13
Electronic Location ID: e19393
Received 2024 Dec 2; Accepted 2025 Apr 8
Copyright: ©2025 Kumrai et al.
Copyright year: 2025
Copyright holder: Kumrai et al.
License: This is an open access article distributed under the terms of the Creative Commons Attribution License, which permits unrestricted use, distribution, reproduction and adaptation in any medium and for any purpose provided that it is properly attributed. For attribution, the original author(s), title, publication source (PeerJ) and either DOI or URL of the article must be cited.
License URL: https://creativecommons.org/licenses/by/4.0/

Keywords: Aneurysm wall characteristics assessment, 4D-CT angiography, CNN-LSTM-based regression model

Funding: JST, CREST Grant Number JPMJCR2013, Japan This work was supported by JST, CREST Grant Number JPMJCR2013, Japan. The funders had no role in study design, data collection and analysis, decision to publish, or preparation of the manuscript.

==============================
Purpose

This study proposes a novel deep learning-based approach for aneurysm wall characteristics, including thin-walled (TW) and hyperplastic-remodeling (HR) regions.

Materials and Methods

We analyzed fifty-two unruptured cerebral aneurysms employing 4D-computed tomography angiography (4D-CTA) and intraoperative recordings. The TW and HR regions were identified in intraoperative images. The 3D trajectories of observation points on aneurysm walls were processed to compute a time series of 3D speed, acceleration, and smoothness of motion, aiming to evaluate the aneurysm wall characteristics. To facilitate point-level risk evaluation using the time-series data, we developed a convolutional neural network (CNN)—long- short-term memory (LSTM)-based regression model enriched with attention layers. In order to accommodate patient heterogeneity, a patient-independent feature extraction mechanism was introduced. Furthermore, unlabeled data were incorporated to enhance the data-intensive deep model.

Results

The proposed method achieved an average diagnostic accuracy of 92%, significantly outperforming a simpler model lacking attention. These results underscore the significance of patient-independent feature extraction and the use of unlabeled data.

Conclusion

This study demonstrates the efficacy of a fine-grained deep learning approach in predicting aneurysm wall characteristics using 4D-CTA. Notably, incorporating an attention-based network structure proved to be particularly effective, contributing to enhanced performance.

Introduction

Subarachnoid hemorrhage resulting from ruptured cerebral aneurysms poses substantial morbidity and mortality risks. A thin-walled (TW) region of an unruptured cerebral aneurysm can be at potential risk of an aneurysm rupture during the natural course. Conversely, atherosclerosis is the result of an hyperplastic-remodeling (HR) region, which may cause an adverse effect on surgical treatment (Int. Study of Unruptured Intracranial Aneurysms, 1998; Wiebers, 2003). Therefore, a preoperative assessment of aneurysm wall characteristics can contribute to the decision-making of the management of unruptured cerebral aneurysms (Cornelissen et al., 2015; Qian et al., 2011; Takao et al., 2012). While 3D-computed tomography angiography (3D-CTA) (Tanioka et al., 2020; White et al., 2001) has traditionally been instrumental in evaluating morphological and anatomical characteristics, 4D-CTA (Ishida et al., 2005) can provide the dynamics of an aneurysm wall relating to aneurysm growth, bleb formation, and rupture point. To capture these dynamics nuances, this study employs deep learning-based supervised machine learning for point-level risk assessment in cerebral aneurysms using 4D-CTA. Here the point-level assessment means assessment for each point of interest on an aneurysm wall. This study faces two primary challenges. The first challenge of this study is the high cost of preparing sufficient labeled training data by referring to intraoperative images. As shown in Fig. 1A, it is straightforward to label apparent cases as a TW or an HR region. However, identifying labels for other ambiguous cases is problematic, resulting in a limited amount of training data. Secondly, collecting labeled data from a target patient is not practical. Ideally, data should be gathered from a large pool of patients, but in this case, the challenge is the inherent differences among patients. In this study, we regard ambiguous difficult-to-label points as unlabeled data and leverage the data to supplement labeled data to train a data-hungry deep model. Additionally, in addressing the issue of patient heterogeneity, we introduce constraints during the model training to ensure the extraction of patient-invariant features.

Figure 1 (A) An example intraoperative image of a cerebral aneurysm. (B) An example 3D trajectory of a point of interest on an aneurysm wall (smoothed).

Materials and Methods

Dataset

This retrospective study included 52 unruptured cerebral aneurysms treated at Asahikawa Red Cross Hospital, Osaka University Hospital, Saiseikai Kumamoto Hospital, and Mie Chuo Medical Center. All patients are Japanese with a gender ratio of 30% male and 70% female. The mean and standard deviation of age are 63.33 and 8.87, respectively. Prior to analysis, patient records underwent a rigorous anonymization process to safeguard privacy.

The 4D-CTA imaging data was acquired using 320-detector CT scanners (Aquilion ONE GENESIS, and Aquilion ONE, Canon Medical Systems Corporation, Otawara, Japan). The collected imaging data was analyzed using a specialized medical image-processing software. The aneurysm-containing region was then precisely identified and extracted. Two key outputs were generated: (1) a 4D-CTA geometry file representing the spatial characteristics of the aneurysm and surrounding structures, and (2) a comprehensive point-trajectory file capturing the dynamic vascular behavior within the region of interest. To analyze vascular motion, we collected the 3D trajectories of each observation point, corresponding to a single pixel in the imaging data on the aneurysm walls, such as middle cerebral artery (MCA; 30 cases), anterior cerebral arteries (ACA; two cases), internal carotid artery—posterior cerebral artery (IC-PC; 14 cases), and anterior communicating artery (A-com; six cases). Voxel tracking was employed for trajectory extraction, and Fig. 1B provides a visual representation of one such trajectory. The trajectories were sampled at a rate of 100 Hz with each trajectory spanning a duration of 1 s. To achieve higher temporal resolution for accurate tracking, a nonlinear interpolation method was applied, with its precision benefits outweighing potential errors. Phase diagrams, derived from the interpolated data and representing displacement, velocity, and acceleration of membrane motion, demonstrated strong consistency with established physical principles. This consistency confirms that enhanced temporal sampling preserves critical biomechanical characteristics while enabling a more detailed analysis of dynamic vascular behaviors.

We annotated each point in aneurysm walls based on RGB images recorded during craniotomies. Note that all surgeries used the same medical microscope model. These microscopes had professional lighting systems with constant illumination, which ensured stable light intensity, color temperature, and uniformity during image capture. As a result, variations from natural or indoor lighting were eliminated. To achieve point-to-point matching between the 4D-CTA data and the labeled points from the RGB image, we implemented a dual-approach point-cloud generation process. First, we created the primary point cloud directly from the original 4D-CTA data. Then, we generated a corresponding point cloud from the RGB images using a pattern-matching approach. This method involved identifying anatomical or vascular structures that were present in both the intraoperative RGB images and the 3D geometry reconstructed from 4D-CTA data. By detecting these shared features and establishing spatial correspondences, we accurately aligned the RGB images with the 3D geometry. This alignment enabled the precise projection of RGB-derived information onto the 4D-CTA geometry file, effectively integrating color and texture details into the 3D representation. Finally, to transfer the color labels to the original point cloud, a nearest-point search was conducted between the primary 4D-CTA point cloud and the newly generated colored point cloud. This comprehensive process resulted in a unified point cloud that seamlessly integrated the color labels and accurately represented the aneurysm’s structural and visual characteristics.

There are 11 annotators with experience ranging from 10 days to 5 years, 6 months, and 24 days. The average expertise is 1 year, 2 months, and 17 days. At least three annotators are involved in a majority vote, and they consistently receive advice from the doctors from the mentioned hospitals throughout the entire annotation process. Additionally, another three to five annotators are engaged in discussions regarding essential parts that require judgment. We performed a thorough verification process including manual review and automated checks to confirm the dataset is complete and free from missing data. We labeled points that exhibited an apparent blue or red coloration, corresponding to thin-walled (TW) or hyperplastic-remodeling (HR) regions, as illustrated in Figs. 2A–2B. Notably, we do not measure the actual thickness for TW and HR; instead, we conduct a relative evaluation as described above. Moreover, the total count of points as TW and HR amounted to 104,000 each. The remaining 312,000 points have no class labels. This study was approved by the Osaka University Research Ethics Committee Institutional Review Board (Reference Number: R 48-7, Approval Number: 201905, Ethical Application Reference: 15000107), and written informed consent was obtained from all participants.

Figure 2 (A) An example RGB image recorded during craniotomy. (B) An example of an RGB image with integrated color labels.

Neural network model

Capturing long- and short-term temporal dynamics plays a pivotal role in point-level assessment. As shown in Fig. 3, we have devised a neural network composed of 1D convolution (CNN) layers and long- short-term memory (LSTM) layers. This architecture enables us to capture short- and long-term trends, respectively. Additionally, we introduced attention layers to focus on important time steps (Vaswani et al., 2017). Note that we employed Glorot uniform initialization (Glorot & Bengio, 2010) for the weights and zero initialization for the biases. The input of the model is a time series of 3D speed, acceleration, and smoothness of motion calculated from a 3D trajectory of a point of interest. The smoothness of motion can be represented by the angles between three adjacent points (pt−1, pt, and pt+1, where pt is the coordinate at time t) in the trajectory. Consistent angles indicate smooth motion, while highly variable angles suggest unstable or jerky movement. In this study, we considered the angles between three adjacent points in 2D planes of the 3D space: XY, XZ, and YZ planes. The output is a numeric value showing the status of the point: 0 for TW or 1 for HR. In this study, we developed and trained the neural network model for classification1 using TensorFlow 2.6.2 with the Keras API in Python 3.6.13. The training was conducted in a GPU environment equipped with a ROG-STRIX-RTX4090-O24G (24 GB) GPU, an Intel Core i9-13900KF (13th Gen, 5.8 GHz) CPU, and 64 GB of Trident Z5 RGB DDR5 6,400 MHz RAM.

Figure 3 Neural network architecture.

Model training

The Adam optimizer (Kingma, 2014) was used to train the network based on the following loss function. (1) Eθf,yi,yi ˆ=1N∑i=1Nyi−yi ˆ2+1N∑i=1NLpθf−1N∑i=1NLmθf

where θf represents network parameters of the feature extraction layer (i.e., Dense_1 layer), and N denotes the number of training instances. The first term calculates the mean absolute error (MAE) between the ground-truth label (yi) and estimate (y ˆi). The second term was introduced to extract patient-independent features and Lp() denotes a loss term for a patient-independent feature that calculates the Euclidean distance between data points from different patients with the same class labels in the feature space (output of dense_1 layer). Minimizing this term ensures that points from different patients with the same labels have similar movement features, i.e., patient-independent features. The third term was introduced to leverage unlabeled data and Lm() denotes a loss term for differences in movement features between labeled and unlabeled data that calculates the Euclidean distance between a labeled point (TW or HR) and an unlabeled point in the feature space. Because the unlabeled points do not belong to TW or HR, this term ensures that the unlabeled points have different movement features from the labeled points. Also, we believe that this term helps us acquire the latent relationship between points in TW and HR regions by leveraging the ambiguous points as a proxy.

Evaluation methodology

To evaluate the effectiveness of the proposed method, we prepared the following methods:

• Proposed: The proposed method.

• Baseline: This method is based on an LSTM-based regression model composed of an LSTM layer and two dense layers. The model is trained by minimizing the loss function that calculates only the MAE.

• Only MAE: This is a variation of the proposed method. This method involves training a CNN-LSTM-based attention model by minimizing the loss function, which calculates only the MAE.

• W/o patient-Ind: This is a variant of the proposed method, but it disables the term for patient-independent feature extraction, i.e., the second term in Eq. (1).

• W/o unlabeled: This is a variant of the proposed method, but it disables the term responsible for leveraging unlabeled data, i.e., the third term in Eq. (1).

The above methods were evaluated on the leave-one-patient-out cross-validation. We used accuracy, precision, recall, and F1-score as evaluation metrics. Note that the output of each method is a numeric value. In this study, we used a threshold of 0.5 to distinguish between TW (0) and HR (1).

Results

In Table 1 (All patients), we present the classification performance of each method, averaged across 52 patients. As depicted in the table, Only MAE significantly outperformed the Baseline, and the Baseline is closely comparable to random classification. W/o patient-Ind and W/o unlabeled outperformed Only MAE by about 2.5% and 3% in F1-score, respectively. Proposed achieved the highest accuracy and outperformed Only MAE by approximately 3.5%. Furthermore, we noticed that Proposed greatly improved the accuracy for patients who failed with Only MAE. Among 52 patients, six of them exhibited an accuracy lower than 80% when employing the Only MAE method. Table 1 (Failed patients) shows the classification results for the failed patients. W/o patient-Ind and W/o unlabeled outperformed Only MAE by about 18% in F1-score. Our proposed method excels with the highest accuracy and exhibits an improvement of around 20.25% in F1-score compared to Only MAE.

Table 1 Average accuracy of point-level risk assessment predictions (%) for cerebral aneurysms.

Methods		Accuracy	Precision	Recall	F1-score	
	All patients (52 patients)					
Proposed		92.94	93.58	92.94	92.88	
Baseline		50.00	25.00	50.00	33.33	
Only MAE		89.93	91.51	89.93	89.42	
W/o patient-Ind		92.05	92.89	92.05	91.86	
W/o unlabeled		92.60	93.30	92.60	92.51	
	** Failed patients (6 patients)					
Proposed		85.59	87.07	85.59	85.44	
Baseline		50.00	25.00	50.00	33.33	
Only MAE		69.15	78.44	69.15	65.19	
W/o patient-Ind		84.40	88.03	84.40	83.12	
W/o unlabeled		84.01	86.09	84.01	83.73	
Notes.

** Failed patients are those with an accuracy lower than 80% when using the Only MAE method.

Discussion

As shown in Table 1 (All patients), Only MAE significantly outperformed the Baseline, showing the significance of the attention-based architecture. The predictions were almost random without using the attention layers, indicating that the attention layers focus on patient-independent features (time segments in input data). Figures 4A–4B shows examples of TW and HR trajectories colored with attention values. Segments corresponding to high-speed large movements have high attention values in these trajectories (specifically in the TW example), indicating characteristic movements of light-mass TW regions. The results of W/o patient-Ind and W/o unlabeled show that the contributions of the loss for patient-independent features and the loss for unlabeled data were almost the same. By combining the two loss terms, Proposed achieved the best performance. Furthermore, Fig. 5 shows a Taylor diagram that quantifies the degree of correspondence between the method and observed behavior using three statistics: the standard deviation (SD), the Pearson correlation coefficient, and the root-mean-square error (RMSE). The Taylor diagram shows that the Proposed method agrees best with observations compared to the other methods because it has the highest correlation, the smallest RMSE, and the SD closest to the reference. Table 1 (Failed patients) shows that the contribution of the loss term for patient-independent features is higher than the loss term for differences in movement features between labeled and unlabeled data because W/o unlabeled outperformed W/o patient-Ind. The result indicates that it is difficult to precisely predict point-level risk assessment in failed patients because the data distribution of failed patients differs from that of the majority.

Figure 4 (A) An example of 3D trajectory of TW colored with an attention value for each time step (average attention value of all attention layers). (B) An example of 3D trajectory of HR colored with an attention value for each time step (average attention value of all attention layers).

Figure 5 Taylor diagram.

To further investigate these distribution differences, we conducted the two-sample Kolmogorov–Smirnov (KS) test (Hodges Jr, 1958) to analyze the distribution of two samples (TW and HR) in terms of speed, acceleration, and smoothness of motion features for each patient and compared the statistics between the failed patients and succeeded patients (those with accuracy higher than 80%), as illustrated in Fig. 6. These results show that the KS statistic for the failed patients is smaller than that for the succeeded patients, indicating a small difference in speed, acceleration, or smoothness of motion between TW and HR regions. Additionally, the KS test yielded a p-value close to zero, indicating significant differences in feature distributions between failed and succeeded patients. Furthermore, we performed Levene’s test (a variance equality test) (Brown & Forsythe, 1974), which produced a p-value close to zero, suggesting greater variability in movement characteristics among failed patients. To further analyze failed patients, we measured dynamic time warping (DTW) distance (Müller, 2007), and Wasserstein distances (Villani & Villani, 2009). The DTW distances for both TW and HR regions ranged from approximately 0.29 to 6.85, indicating erratic and highly variable movement patterns. Similarly, Wasserstein distances ranged from 0.0028 to 0.0257, reflecting significant discrepancies in trajectory distributions. These extreme values suggest that failed patients not only deviate from the majority distribution, but also exhibit irregular and unpredictable motion behaviors. However, the proposed method, which employs the two loss terms, could improve the F1-score of these failed patients by about 20.25% compared to Only MAE. This study leveraged deep learning-based supervised machine learning for point-level risk assessment in cerebral aneurysms using a time series of 3D speed, acceleration, and smoothness of motion data derived from 4D-CTA. To ensure robust model evaluation, we evaluated our proposed method on the leave-one-patient-out cross-validation, where each patient serves as a test case while the model is trained on the remaining data. As demonstrated, our proposed method effectively predicted point-level risk labels without requiring labeled data specific to a target patient. These results will contribute to predicting aneurysm wall characteristics for decision-making of unruptured cerebral aneurysms. Although the number of patients in our study is relatively small (N = 52), each patient contributed a substantial number of data points (2,000 trajectories per patient), resulting in a total dataset of 104,000 samples. This large dataset enables the model to learn meaningful patterns despite the limited patient count. However, expanding the dataset remains a key objective for future work to further enhance model performance and reliability.

Figure 6 Boxplot of the two-sample Kolmogorov–Smirnov test to compare speed, acceleration, and smoothness of motion of TW with HR regions for each patient.

Conclusions

A deep learning-based supervised machine learning approach was proposed for point-level risk assessment in cerebral aneurysms using time series data of 3D speed, acceleration, and smoothness of motion, all derived from 4D-CTA. The introduced patient-independent feature extraction mechanism can accommodate patient heterogeneity, while the use of unlabeled data can enhance the data-intensive deep model. A time series of 3D speed, acceleration, and smoothness of motion data derived from 4D-CTA can be used to train deep learning-based supervised machine learning for aneurysm wall characteristics assessment. The point-level risk labels can be predicted without using labeled data specific to a target patient. However, the prediction might fail in cases where the data distribution of the target patient differs from that of the majority.

Supplemental Information

Supplemental Information 1 Python Code

Additional Information and Declarations

Competing Interests

Author Contributions

Human Ethics

Data Availability

1 The source code used in this study is available on GitHub: https://github.com/Kumrai-T/DA_4DCTA.

The authors declare there are no competing interests.

Teerawat Kumrai conceived and designed the experiments, performed the experiments, prepared figures and/or tables, authored or reviewed drafts of the article, and approved the final draft.

Takuya Maekawa conceived and designed the experiments, performed the experiments, prepared figures and/or tables, authored or reviewed drafts of the article, and approved the final draft.

Yixuan Chen analyzed the data, authored or reviewed drafts of the article, and approved the final draft.

Yoshie Sugiyama conceived and designed the experiments, performed the experiments, analyzed the data, authored or reviewed drafts of the article, and approved the final draft.

Masatoshi Takagaki analyzed the data, authored or reviewed drafts of the article, and approved the final draft.

Shigeo Yamashiro analyzed the data, authored or reviewed drafts of the article, and approved the final draft.

Katsumi Takizawa analyzed the data, authored or reviewed drafts of the article, and approved the final draft.

Tsutomu Ichinose analyzed the data, authored or reviewed drafts of the article, and approved the final draft.

Fujimaro Ishida analyzed the data, authored or reviewed drafts of the article, and approved the final draft.

Haruhiko Kishima analyzed the data, authored or reviewed drafts of the article, and approved the final draft.

The following information was supplied relating to ethical approvals (i.e., approving body and any reference numbers):

The Osaka University Research Ethics Committee granted ethical approval to conduct the study within its facilities (Reference Number: R 48-7, Ethical Application Ref: 15000107).

The following information was supplied regarding data availability:

The source code and raw data are available at GitHub and Zenodo

- https://github.com/Kumrai-T/DA_4DCTA

- Teerawat Kumrai. (2024). Kumrai-T/DA_4DCTA: Deep learning-based approach for aneurysm wall characteristics using 4DCTA (v1.0.1). Zenodo. https://doi.org/10.5281/zenodo.13788524.

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
