# Peer review of "Deep learning-based fine-grained assessment of aneurysm wall characteristics using 4D-CT angiography"

_PeerJ, doi:10.7717/peerj.19393_

## Round 0.1 · original submission · Major Revisions

The reviewers recognise this study's innovation and potential clinical applications. They highlight the strong performance of you deep learning model in assessing aneurysm wall characteristics. However, they raise several methodological concerns that need addressing to strengthen your manuscript. Key issues that they raise include unclear details on matching intraoperative tumor images with 4D-CTA images, insufficient description of the CNN-LSTM model construction, and a lack of information on 4D-CTA imaging parameters and registration methods. They also raise concerns about the annotation process, such as potential environmental variations and the need for validation metrics. Additionally, the 100Hz sampling rate for tracking trajectory points lacks clarity on interpolation or oversampling errors. A major limitation, which I currently share, is the absence of external validation, raising questions about the model's generalisability. The reviewers suggest discussing failed cases, but they also recommend clearly justifying the sample size of 52 patients. Overall, while your study is certainly promising, if you are to address these concerns I am sure it will improve the manuscript's transparency, validity and robustness.

Reviewer 1 ·

Basic reporting

This manuscript titled“Deep Learning-Based Fine-Grained Assessment of Aneurysm Wall Characteristics Using 4D-CT Angiography” uses a deep learning-based supervised machine learning approach to overcome the challenges of small sample size and large degree of variability in the training set. By matching the direct view photos of aneurysms in the surgical field with the 4D-CTA data, and tracking the movement trajectory changes of points of interest, the study constructed a convolutional neural network (CNN) - long- short-term memory (LSTM)-based regression model to predict thin-walled (TW) and hyperplastic-remodeling (HR) regions of aneurysms. The model performance evaluation indicates that the established model has good disrimination and accuracy. The research is innovative, using artificial intelligence algorithms digging useful information of 4D-CTA data to evaluate risk point of an aneurysm, which do have clinical application potential.
The English writing is clear and unambiguous, relevant prior literature is cited appropriate.

Experimental design

However, there are some issues that need to be addressed:
1、 The MATERIALS AND METHODS section, I cannot find how to match the tumor images obtained during the operation with the images of 4D-CTA, using what method, a software or a self-created program? Please try to explain it.
2、 The program used to construction the convolutional neural network (CNN) - long- short-term memory (LSTM)-based regression model is not introduced in the MATERIALS AND METHODS section. May you add it to this part
3、 The basic parameters of 4D-CTA, especially spatial resolution and temporal resolution, and also the size of the predicted TW and HR point should be strengthened and give some explanation.
4、 There is no external data set for validation, which is an imperfect feature of this study. I suggest that the authors give some statement about this in the Discussion section.

Validity of the findings

no comment

Additional comments

no comment

Annotated reviews are not available for download in order to protect the identity of reviewers who chose to remain anonymous.

Reviewer 2 ·

Basic reporting

The authors use deep learning to assess aneurysm wall characteristics using 4D-CT angiography. The research highlights the potential of advanced computational methods for improving risk assessment and clinical decision-making.

Experimental design

1. The observation point is not clear. Are these points single pixels or areas encompassing multiple pixels?
2. The annotation process appears to rely on recorded images, which could be influenced by environmental changes across different cases. Additionally, significant variability in annotator experience is noted. Could you provide evidence or metrics to validate the accuracy and consistency of the annotated data? Including sample annotated images would also help demonstrate the quality of your dataset.
3. The manuscript mentions sampling trajectory points at 100Hz but does not specify any interpolation method used. Furthermore, are there any potential errors introduced by over-sampling?
4. The details regarding the 4D-CTA data processing are limited. Could you provide more information about the imaging data specifications and how the raw data were processed?
5. What method was used to register the point-to-point between the data of 4D-CTA and the labelled points from the RGB image?

Validity of the findings

1. The model was evaluated using the leave-one-patient-out cross-validation method. Whether the model has been validated on external datasets or whether there are plans for future validation studies?
2. More justification and discussion are expected of the failed patients' cases.
3. The authors should add more discussion about whether 52 patients are sufficient to accurately train the proposed model.

Additional comments

Overall, it is an interesting topic. The manuscript could be strengthened by providing more study details and discussions.

---

## Round 0.2 · accepted · Accept

All comments have been addressed, and this paper is now ready for publication

Reviewer 2 ·

Basic reporting

The authors have answered my raised questions.

Experimental design

The authors have provided more information regarding the comments.

Validity of the findings

No more comments.